# Inception Transformer

Chenyang Si[1*] Weihao Yu[1,2*] Pan Zhou[1] Yichen Zhou[1,2] Xinchao Wang[2] Shuicheng Yan[1]

[1]Sea AI Lab    [2]National University of Singapore
{sicy,yuweihao,zhoupan,zhouyc,yansc}@sea.com, xinchao@nus.edu.sg

## Abstract

Recent studies show that Transformer has strong capability of building long-range dependencies, yet is incompetent in capturing high frequencies that predominantly convey local information. To tackle this issue, we present a novel and general-purpose *Inception Transformer*, or *iFormer* for short, that effectively learns comprehensive features with both high- and low-frequency information in visual data. Specifically, we design an Inception mixer to explicitly graft the advantages of convolution and max-pooling for capturing the high-frequency information to Transformers. Different from recent hybrid frameworks, the Inception mixer brings greater efficiency through a channel splitting mechanism to adopt parallel convolution/max-pooling path and self-attention path as high- and low-frequency mixers, while having the flexibility to model discriminative information scattered within a wide frequency range. Considering that bottom layers play more roles in capturing high-frequency details while top layers more in modeling low-frequency global information, we further introduce a frequency ramp structure, *i.e.*, gradually decreasing the dimensions fed to the high-frequency mixer and increasing those to the low-frequency mixer, which can effectively trade-off high- and low-frequency components across different layers. We benchmark the iFormer on a series of vision tasks, and showcase that it achieves impressive performance on image classification, COCO detection and ADE20K segmentation. For example, our iFormer-S hits the top-1 accuracy of $83.4\%$ on ImageNet-1K, much higher than DeiT-S by $3.6\%$, and even slightly better than much bigger model Swin-B ($83.3\%$) with only 1/4 parameters and 1/3 FLOPs. Code and models are released at https://github.com/sail-sg/iFormer.

## 1 Introduction

Transformer [1] has taken the natural language processing (NLP) domain by storm, achieving surprisingly high performance in many NLP tasks, *e.g.*, machine translation [2] and question-answering [3]. This is largely attributed to its strong capability of modeling long-range dependencies in the data with self-attention mechanism. Its success has led researchers to investigate its adaptation to the computer vision field, and Vision Transformer (ViT) [4] is a pioneer. This architecture is directly inherited from NLP [1], but applied to image classification with raw image patches as input. Later, many ViT variants [5–13] have been developed to boost performance or scale to a wider range of vision tasks, *e.g.*, object detection [10, 11] and segmentation [12, 13].

ViT and its variants are highly capable of capturing low-frequencies in the visual data [14], mainly including global shapes and structures of a scene or object, but are not very powerful for learning high-frequencies, mainly including local edges and textures. This can be intuitively explained: self-attention, the main operation used in ViTs to exchange information among non-overlap patch tokens, is a global operation and much more capable of capturing global information (low frequencies) in the

---

*Equal contribution. Weihao Yu did this work during an internship at Sea AI Lab.

36th Conference on Neural Information Processing Systems (NeurIPS 2022).

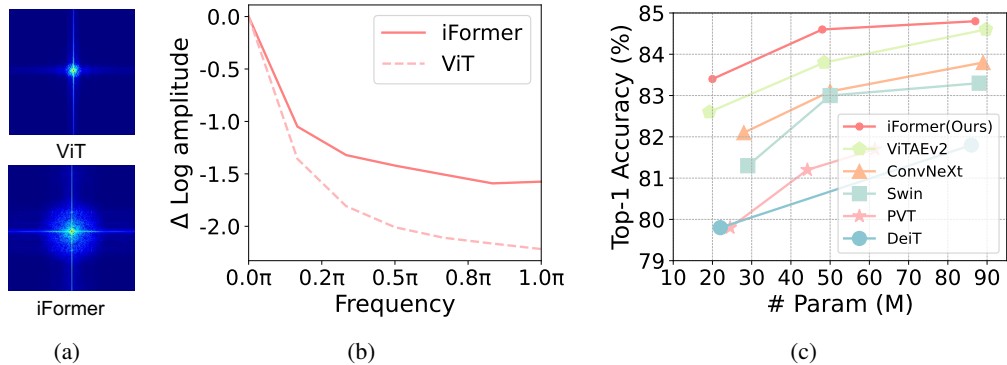

Figure 1: **(a) Fourier spectrum of ViT [18] and iFormer. (b) Relative log amplitudes of Fourier transformed feature maps. (c) Performance of models on ImageNet-1K validation set.** (a) and (b) show that iFormer captures more high-frequency signals.

data than local information (high frequencies). As shown in Fig. 1(a) and 1(b), the Fourier spectrum and relative log amplitudes of the Fourier show that ViT tends to well capture low-frequency signals but few high-frequency signals. This observation also accords with the empirical results in [14], which shows ViT presents the characteristics of low-pass filters. This low-frequency preferability impairs the performance of ViTs, as 1) low-frequency information filling in all the layers may deteriorate high-frequency components, *e.g.*, local textures, and weakens modeling capability of ViTs; 2) high-frequency information is also discriminative and can benefit many tasks, *e.g.*, (fine-grained) classification. Actually, human visual system extracts visual elementary features at different frequencies [15–17]: low frequency provides global information about a visual stimulus, and high frequency conveys local spatial changes in the image (*e.g.*, local edges/textures). Hence, it is necessary to develop a new ViT architecture for capturing both high and low frequencies in the visual data.

CNNs are the most fundamental backbone for general vision tasks. Unlike ViTs, they cover more local information through local convolution within the receptive fields, thus effectively extracting high-frequency representations [19, 20]. Recent studies [21–25] have integrated CNNs and ViTs considering their complementary advantages. Some methods [21, 22, 24, 25] stack convolution and attention layers in a serial manner to inject the local information into global context. Unfortunately, this serial manner only models one type of dependency, either global or local, in one layer, and discards the global information during locality modeling, or vice versa. Other works [23, 26] adopt parallel attention and convolution to learn global and local dependencies of the input at the same time. However, it is found in [27] that part of the channels are for processing local information and the other for global modeling, meaning current parallel structures have information redundancy if processing all channels in each branch.

To address this issue, we propose a simple and efficient *Inception Transformer (iFormer)*, as shown in Fig. 2, which grafts the merit of CNNs for capturing high-frequencies to ViTs. The key component in iFormer is an Inception token mixer as shown in Fig. 3. This Inception mixer aims to augment the perception capability of ViTs in the frequency spectrum by capturing both high and low frequencies in the data. To this end, the Inception mixer first splits the input feature along the channel dimension, and then feeds the split components into high-frequency mixer and low-frequency mixer respectively. Here the high-frequency mixer consists of a max-pooling operation and a parallel convolution operation, while the low-frequency mixer is implemented by a vanilla self-attention in ViTs. In this way, our iFormer can effectively capture particular frequency information on the corresponding channel, and thus learn more comprehensive features within a wide frequency range compared with vanilla ViTs, which can be clearly observed in Fig. 1(a) and 1(b).

Moreover, we find that lower layers often need more local information, while higher layers desire more global information, which also accords with the observations in [27]. This is because, like in human visual system, the details in high frequency components help lower layers to capture visual elementary features and also to gradually gather local information for having a global understanding of the input. Inspired by this, we design a frequency ramp structure. In particular, from lower to higher layers, we gradually feed more channel dimensions to low-frequency mixer and fewer channel

dimensions to high-frequency mixer. This structure can trade-off high-frequency and low-frequency components across all layers. Its effectiveness has been verified by experimental results in Sec. 4.

Experimental results show that iFormer surpasses state-of-the-art ViTs and CNNs on several vision tasks, including image classification, object detection and segmentation. For example, as shown in Fig. 1(c), with different model sizes, iFormer makes consistent improvements over popular frameworks on ImageNet-1K [28], *e.g.*, DeiT [29], Swin [5] and ConvNeXt [30]. Meanwhile, iFormer outperforms recent frameworks on COCO [31] detection and ADE20K [32] segmentation.

## 2 Related work

Transformers [1] are firstly proposed for machine translation tasks and then become popular in other tasks like natural language understanding [33–35] and generation [36, 37] in NLP domain, as well as image classification [18, 29, 38], object detection [6, 39, 40] and semantic segmentation [41, 42] in computer vision. The attention module in Transformers has an outstanding ability to capture global dependency, but it makes the models produce similar representations across layers [27]. Moreover, self-attention mainly captures low-frequency information and tends to neglect high-frequency components related to the detailed information [14].

CNNs [43–47] are the de-facto model for vision tasks due to their outstanding ability to model local dependency [47–49] as well as extract high-frequency [19, 50]. With these advantages, CNNs are rapidly introduced into Transformers in a serial or parallel manner [23–26, 51–53]. For serial methods, convolutions are applied at different positions of the Transformer. CvT [25] and PVT-v2 [54] replace the hard patch embedding with a layer of overlapping convolution. LV-ViT [51], LeViT [55] and ViT$_C$ [21] further stack several layers of convolutions as the stem for models, which is found helpful in training and achieving better performance. Besides the stem, ViT-hybrid [18], CoAtNet [24], Hybrid-MS [56] and UniFormer [22] design early stages with convolution layers. However, the combination of convolution and attention in a serial order means each layer can only process either high or low frequency and neglects the other part. To enable each layer to process different frequencies, we adopt the parallel manner to combine convolution and attention in a token mixer.

Compared with serial methods, there are not many works combining attention and convolution in a parallel manner in literature. CoaT [26] and ViTAE [23] introduce convolution as a branch parallel to attention and utilize elementwise sum to merge the output of the two branches. However, Raghu *et al.* find that some channels tend to extract local dependency while others are for modeling global information [27], indicating redundancy for the current parallel mechanism to process all channels in different branches. In contrast, we split channels into branches of high and low frequencies. GLiT [53] also adopt parallel manner but it directly concatenate the features from convolution and attention branches as the mixer output, lacking the fusion of features in different frequencies. Instead, we design a explicit fusion module to merge the outputs from low- and high-frequency branches.

## 3 Method

### 3.1 Revisit Vision Transformer

We first revisit the Vision Transformer. For vision tasks, Transformers first split the input image into a sequence of tokens, and each patch token is projected into a hidden representation vector with a leaner layer, denoted as $\{\boldsymbol{x}_1, \boldsymbol{x}_2, ..., \boldsymbol{x}_N\}$ or $\boldsymbol{X} \in \mathbb{R}^{N \times C}$, where $N$ is the number of patch tokens and $C$ indicates the dimension of features. Then, all of the tokens are combined with a positional embedding and fed into the Transformer layers that contain multi-head self-attention (MSA) and a feed-forward network (FFN).

In MSA, the attention-based mixer exchanges information between all patch tokens so that it strongly focuses on aggregating the global dependency across all layers. However, excessive propagation of global information would strengthen the low-frequency representation. It can be seen from the visualization of Fourier spectrum in Fig. 1(a) that low-frequency information dominates the representations of ViT [18]. This actually impairs the performance of ViTs, as it may deteriorate the high-frequency components, *e.g.*, local textures, and weakens the modeling capability of ViTs [14]. In the visual data, high-frequency information is also discriminative and can benefit many tasks

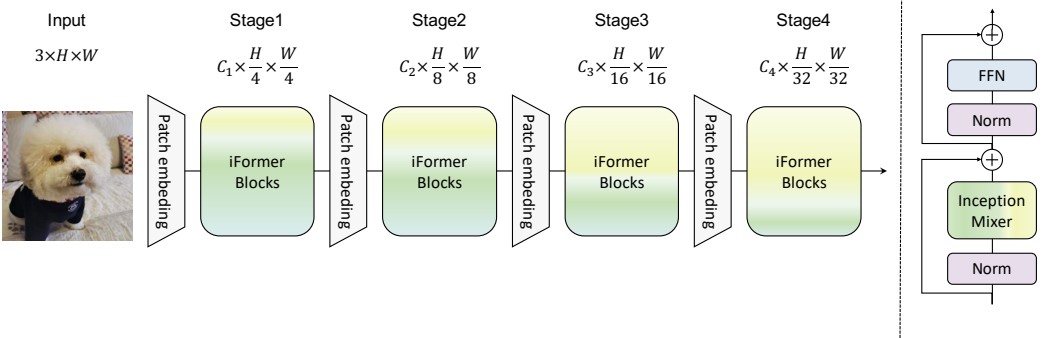

Figure 2: **The overall architecture of iFormer and details of iFormer block** . For each block, yellow and green indicate low- and high-frequency information, respectively. Best viewed in color.

[19, 20]. Hence, to address the issue, we propose a simple and efficient Inception Transformer, as shown in Fig. 2, with two key novelties, *i.e.*, Inception mixer and frequency ramp structure.

## 3.2 Inception token mixer

We propose an Inception mixer to graft the powerful capability of CNNs for extracting high-frequency representation to Transformers. Its detailed architecture is depicted in Fig. 3. We use the name of "Inception" since the token mixer is highly inspired by the Inception module [46, 57–59] with multiple branches. Instead of directly feeding image tokens into the MSA mixer, the Inception mixer first splits the input feature along the channel dimension, and then respectively feeds the split components into high-frequency mixer and low-frequency mixer. Here the high-frequency mixer consists of a max-pooling operation and a parallel convolution operation, while the low-frequency mixer is implemented by a self-attention.

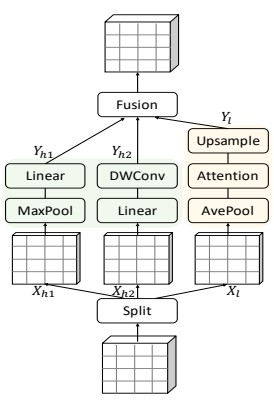

Technically, given the input feature map $X \in \mathbb{R}^{N \times C}$, it is factorized $X$ into $X_h \in \mathbb{R}^{N \times C_h}$ and $X_l \in \mathbb{R}^{N \times C_l}$ along the channel dimension, where $C_h + C_l = C$. Then, $X_h$ and $X_l$ are assigned to high-frequency mixer and low-frequency mixer respectively.

Figure 3: **The details of Inception mixer.**

**High-frequency mixer.** Considering the sharp sensitiveness of the maximum filter and the detail perception of convolution operation, we propose a parallel structure to learn the high-frequency components. We divide the input $X_h$ into $X_{h1} \in \mathbb{R}^{N \times \frac{C_h}{2}}$ and $X_{h2} \in \mathbb{R}^{N \times \frac{C_h}{2}}$ along the channel. As shown in Fig. 3, $X_{h1}$ is embedded with a max-pooling and a linear layer [46], and $X_{h2}$ is fed into a linear and a depthwise convolution layer [60–62]:

$$Y_{h1} = \text{FC}\left(\text{MaxPool}\left(X_{h1}\right)\right), \tag{1}$$
$$Y_{h2} = \text{DwConv}\left(\text{FC}\left(X_{h2}\right)\right), \tag{2}$$

where $Y_{h1}$ and $Y_{h2}$ denote the outputs of high-frequency mixers.

Finally, the outputs of low- and high-frequency mixers are concatenated along the channel dimension:

$$Y_c = \text{Concat}\left(Y_l, Y_{h1}, Y_{h2}\right). \tag{3}$$

The upsample operation in Eq. (7) selects the value of the nearest point for each position to be interpolated regardless of any other points, which results in excessive smoothness between adjacent tokens. We design a fusion module to elegantly overcome this issue, *i.e.*, a depthwise convolution exchanging information between patches, while keeping a cross-channel linear layer that works per location like in previous Transformers. The final output can be expressed as

$$Y = \text{FC}\left(Y_c + \text{DwConv}\left(Y_c\right)\right). \tag{4}$$

Like the vanilla Transformer, our iFormer is equipped with a feed-forward network (FFN), and differently it also incorporates the above Inception token mixer (ITM); LayerNorm (LN) is applied before ITM and FFN. Hence the Inception Transformer block is formally defined as

$$\boldsymbol{Y} = \boldsymbol{X} + \text{ITM}\left(\text{LN}\left(\boldsymbol{X}\right)\right), \tag{5}$$

$$\boldsymbol{H} = \boldsymbol{Y} + \text{FFN}\left(\text{LN}\left(\boldsymbol{Y}\right)\right). \tag{6}$$

**Low-frequency mixer.** We use the vanilla multi-head self-attention to communicate information among all tokens for the low-frequency mixer. Despite the strong capability of the attention for learning global representation, the large resolution of feature maps would bring large computation cost in lower layers. We therefore simply utilize an average pooling layer to reduce the spatial scale of $\boldsymbol{X}_l$ before the attention operation and an upsample layer to recover the original spatial dimension after the attention. This design largely reduces the computational overhead and makes the attention operation focus on embedding global information. This branch can be defined as

$$\boldsymbol{Y}_l = \text{Upsample}\left(\text{MSA}\left(\text{AvePooling}\left(\boldsymbol{X}_l\right)\right)\right), \tag{7}$$

where $\boldsymbol{Y}_l$ is the output of low-frequency mixer. Note that the kernel size and stride for the pooling and upsample layers are set to 2 only at the first two stages.

## 3.3 Frequency ramp structure

In the general visual frameworks, bottom layers play more roles in capturing high-frequency details while top layers more in modeling low-frequency global information, *i.e.*, the hierarchical representations of ResNet [47]. Like humans, by capturing the details in high frequency components, lower layers can capture visual elementary features, and also gradually gather local information to achieve a global understanding of the input. We are inspired to design a frequency ramp structure which gradually splits more channel dimensions from lower to higher layers to low-frequency mixer and thus leave fewer channel dimensions to high-frequency mixer. Specifically, as shown in Fig. 2, our backbone has four stages with different channel and spatial dimensions. For each blocks, we define a channel ratio to better balance the high-frequency and low frequency components, *i.e.*, $\frac{C_h}{C}$ and $\frac{C_l}{C}$, where $\frac{C_h}{C} + \frac{C_l}{C} = 1$. In the proposed frequency ramp structure, $\frac{C_h}{C}$ gradually decreases from shallow to deep layers, while $\frac{C_l}{C}$ gradually increases. Hence, with the flexible frequency ramp structure, iFormer can effectively trade-off high- and low-frequency components across all layers. The configuration of different iFormer models will be described in the appendix.

## 4 Experiments

We evaluate our iFormer on several vision benchmark tasks, *i.e.*, image classification, object detection and semantic segmentation, by comparing it with representative ViTs, CNNs and their hybrid variants. Ablation analysis is also conducted to show the contribution of each novelty in our method. More results will be reported in the appendix.

### 4.1 Results on image classification

**Setup.** For image classification, we evaluate iFormer on the ImageNet dataset [28]. We train the iFormer model with the standard procedure in [6, 22, 29]. Specifically, we use AdamW optimizer with an initial learning rate $1 \times 10^{-3}$ via cosine decay [70], a momentum of 0.9, and a weight decay of 0.05. We set the training epoch number as 300 and the input size as $224 \times 224$. We adopt the same data augmentations and regularization methods in DeiT [29] for fair comparison.

We also use LayerScale [71] to train deep models. Like previous studies [5, 67], we further fine tune iFormer on the input size of $384 \times 384$, with the weight decay of $1 \times 10^{-8}$, learning rate of $1 \times 10^{-5}$, batch size of 512. For fairness, we adopt Timm [72] to implement and train iFormer.

**Results.** Table 1 summarizes the image classification accuracy of all compared methods on ImageNet. For the small model size (∼20M), our iFormer surpasses both the SoTA ViTs and hybrid ViTs, although some ViTs, *e.g.*, Swin [5], Focal [64] and CSwin [65], actually already introduce convolution-like inductive bias into their architectures, and hybrid ViTs directly integrate convolution into ViTs. Specifically, our iFormer-S respectively gains 0.7% and 0.5% top-1 accuracy advantage over SoTA

Table 1: Comparison of different types of models on ImageNet-1K [28].

| Model Size | Arch. | Method | #Param. (M) | FLOPs (G) | Input Size Train | Input Size Test | ImageNet Top-1 | ImageNet Top-5 |
|---|---|---|---|---|---|---|---|---|
| small model size (~20M) | CNN | RSB-ResNet-50 [47, 63] | 26 | 4.1 | 224 | 224 | 80.4 | - |
| | | ConvNeXt-T [30] | 28 | 4.5 | 224 | 224 | 82.1 | - |
| | ViT | Deit-S [29] | 22 | 4.6 | 224 | 224 | 79.8 | 95.0 |
| | | PVT-S [6] | 25 | 3.8 | 224 | 224 | 79.8 | - |
| | | T2T-14 [38] | 22 | 5.2 | 224 | 224 | 80.7 | - |
| | | Swin-T [5] | 29 | 4.5 | 224 | 224 | 81.3 | 95.5 |
| | | Focal-T [64] | 29 | 4.9 | 224 | 224 | 82.2 | 95.9 |
| | | CSwin-T [65] | 23 | 4.3 | 224 | 224 | 82.7 | - |
| | Hybrid | CvT-13 [25] | 20 | 4.5 | 224 | 224 | 81.6 | - |
| | | CoAtNet-0 [24] | 25 | 4.2 | 224 | 224 | 81.6 | - |
| | | Container [66] | 22 | 8.1 | 224 | 224 | 82.7 | - |
| | | ViTAE-S [23] | 24 | 5.6 | 224 | 224 | 82.0 | 95.9 |
| | | ViTAEv2-S [67] | 19 | 5.7 | 224 | 224 | 82.6 | 96.2 |
| | | UniFormer-S [22] | 22 | 3.6 | 224 | 224 | 82.9 | - |
| | | **iFormer-S** | **20** | **4.8** | **224** | **224** | **83.4** | **96.6** |
| medium model size (~50M) | CNN | RSB-ResNet-101 [47, 63] | 45 | 7.9 | 224 | 224 | 81.5 | - |
| | | RSB-ResNet-152 [47, 63] | 60 | 11.6 | 224 | 224 | 82.0 | - |
| | | ConvNeXt-S [30] | 50 | 8.7 | 224 | 224 | 83.1 | - |
| | ViT | PVT-L [6] | 61 | 9.8 | 224 | 224 | 81.7 | - |
| | | T2T-24 [38] | 64 | 13.2 | 224 | 224 | 82.2 | - |
| | | Swin-S [5] | 50 | 8.7 | 224 | 224 | 83.0 | 96.2 |
| | | Focal-S [64] | 51 | 9.1 | 224 | 224 | 83.5 | 96.2 |
| | | CSwin-S [65] | 35 | 6.9 | 224 | 224 | 83.6 | - |
| | Hybrid | CvT-21 [25] | 32 | 7.1 | 224 | 224 | 82.5 | - |
| | | CoAtNet-1 [24] | 42 | 8.4 | 224 | 224 | 83.3 | - |
| | | ViTAEv2-48M [67] | 49 | 13.3 | 224 | 224 | 83.8 | 96.6 |
| | | UniFormer-B [22] | 50 | 8.3 | 224 | 224 | 83.9 | - |
| | | **iFormer-B** | **48** | **9.4** | **224** | **224** | **84.6** | **97.0** |
| large model size (~100M) | CNN | RegNetY-16GF [29, 68] | 84 | 16.0 | 224 | 224 | 82.9 | - |
| | | ConvNeXt-B [30] | 89 | 15.4 | 224 | 224 | 83.8 | - |
| | ViT | DeiT-B [29] | 86 | 17.5 | 224 | 224 | 81.8 | 95.6 |
| | | Swin-B [5] | 88 | 15.4 | 224 | 224 | 83.3 | 96.5 |
| | | Focal-B [64] | 90 | 16.0 | 224 | 224 | 83.8 | 96.5 |
| | | CSwin-B [65] | 78 | 15.0 | 224 | 224 | 84.2 | - |
| | Hybrid | BoTNet-T7 [69] | 79 | 19.3 | 256 | 256 | 84.2 | - |
| | | CoAtNet-3 [24] | 168 | 34.7 | 224 | 224 | 84.5 | - |
| | | ViTAEv2-B [67] | 90 | 24.3 | 224 | 224 | 84.6 | 96.9 |
| | | **iFormer-L** | **87** | **14.0** | **224** | **224** | **84.8** | **97.0** |

ViTs ( *i.e.*, CSwin-T) and hybrid ViTs ( *i.e.*, UniFormer-S), while enjoying the same or smaller model size.

For the medium model size (∼50M), iFormer-B achieves 84.6% top-1 accuracy, and improves over the SoTA ViTs and hybrid ViTs with similar model sizes by significant margins 1.0% and 0.7% respectively. For CNNs, similar to comparison results on medium model size, our iFormer-B outperforms ConvNeXt-S by 1.5%. As for the large mode (∼100M), one can observe similar results on small and medium model sizes.

Table 2 reports the fine-tuning accuracy on the larger resolution, *i.e.*, 384 × 384. One can observe that iFormer consistently outperforms the counterparts by a significant margin across different computation settings. These results clearly demonstrate the advantages of iFormer on image classifications.

Table 2: Fine-tuning Results with larger resolution (384 × 384) on ImageNet-1K [28]. The models in gray color are trained with larger input size.

| Method | #Param. (M) | FLOPs (G) | Input Size Train | Test | ImageNet Top-1 |
|--------|-------------|-----------|------------------|------|----------------|
| EfficientNet-B5 [73] | 30 | 9.9 | 456 | 456 | 83.6 |
| EfficientNetV2-S [74] | 22 | 8.5 | 384 | 384 | 83.9 |
| CSwin-T↑384 [65] | 23 | 14.0 | 224 | 384 | 84.3 |
| CvT-13↑384 [25] | 20 | 16.3 | 224 | 384 | 83.0 |
| CoAtNet-0↑384 [24] | 20 | 13.4 | 224 | 384 | 83.9 |
| ViTAEv2-S↑384 [67] | 19 | 17.8 | 224 | 384 | 83.8 |
| **iFormer-S↑384** | **20** | **16.1** | **224** | **384** | **84.6** |
| EfficientNet-B7 [73] | 66 | 39.2 | 600 | 600 | 84.3 |
| EfficientNetV2-M [74] | 54 | 25.0 | 480 | 480 | 85.1 |
| ViTAEv2-48M ↑384 [67] | 49 | 41.1 | 224 | 384 | 84.7 |
| CSwin-S↑384 [65] | 35 | 22.0 | 224 | 384 | 85.0 |
| CoAtNet-1↑384 [24] | 42 | 27.4 | 224 | 384 | 85.1 |
| **iFormer-B↑384** | **48** | **30.5** | **224** | **384** | **85.7** |
| EfficientNetV2-L [74] | 121 | 53 | 480 | 480 | 85.7 |
| Swin-B↑384 [5] | 88 | 47.0 | 224 | 384 | 84.2 |
| CSwin-B↑384 [65] | 78 | 47.0 | 224 | 384 | 85.4 |
| ViTAEv2-B↑384 [67] | 90 | 74.4 | 224 | 384 | 85.3 |
| CoAtNet-2↑384 [24] | 75 | 49.8 | 224 | 384 | 85.7 |
| **iFormer-L↑384** | **87** | **45.3** | **224** | **384** | **85.8** |

Table 3: Performance of object detection and instance segmentation on COCO val2017 [31]. $AP^b$ and $AP^m$ represent bounding box AP and mask AP, respectively. All models are based on Mask R-CNN [75] and trained by $1\times$ training schedule. The FLOPs are measured at resolution $800\times1280$.

| Method | #Param. (M) | FLOPs (G) | Mask R-CNN 1 × | | | | | |
|--------|-------------|-----------|----------|-------------|-------------|----------|-------------|-------------|
| | | | $AP^b$ | $AP^b_{50}$ | $AP^b_{70}$ | $AP^m$ | $AP^m_{50}$ | $AP^m_{75}$ |
| ResNet50 [47] | 44 | 260 | 38.0 | 58.6 | 41.4 | 34.4 | 55.1 | 36.7 |
| PVT-S [6] | 44 | 245 | 40.4 | 62.9 | 43.8 | 37.8 | 60.1 | 40.3 |
| TwinsP-S [76] | 44 | 245 | 42.9 | 65.8 | 47.1 | 40.0 | 62.7 | 42.9 |
| Twins-S [76] | 44 | 228 | 43.4 | 66.0 | 47.3 | 40.3 | 63.2 | 43.4 |
| Swin-T [5] | 48 | 264 | 42.2 | 64.6 | 46.2 | 39.1 | 61.6 | 42.0 |
| ViL-S [77] | 45 | 218 | 44.9 | 67.1 | 49.3 | 41.0 | 64.2 | 44.1 |
| Focal-T [64] | 49 | 291 | 44.8 | 67.7 | 49.2 | 41.0 | 64.7 | 44.2 |
| UniFormer-S$_{h14}$ [22] | 41 | 269 | 45.6 | 68.1 | 49.7 | 41.6 | 64.8 | 45.0 |
| **iFormer-S** | **40** | **263** | **46.2** | **68.5** | **50.6** | **41.9** | **65.3** | **45.0** |
| ResNet101 [47] | 63 | 336 | 40.4 | 61.1 | 44.2 | 36.4 | 57.7 | 38.8 |
| X101-32 | 63 | 340 | 41.9 | 62.5 | 45.9 | 37.5 | 59.4 | 40.2 |
| PVT-M [6] | 64 | 302 | 42.0 | 64.4 | 45.6 | 39.0 | 61.6 | 42.1 |
| TwinsP-B [76] | 64 | 302 | 44.6 | 66.7 | 48.9 | 40.9 | 63.8 | 44.2 |
| Twins-B [76] | 76 | 340 | 45.2 | 67.6 | 49.3 | 41.5 | 64.5 | 44.8 |
| Swin-S [5] | 69 | 354 | 44.8 | 66.6 | 48.9 | 40.9 | 63.4 | 44.2 |
| Focal-S [64] | 71 | 401 | 47.4 | 69.8 | 51.9 | 42.8 | 66.6 | 46.1 |
| CSWin-S [65] | 54 | 342 | 47.9 | 70.1 | 52.6 | 43.2 | 67.1 | 46.2 |
| UniFormer-B [22] | 69 | 399 | 47.4 | 69.7 | 52.1 | 43.1 | 66.0 | 46.5 |
| **iFormer-B** | **67** | **351** | **48.3** | **70.3** | **53.2** | **43.4** | **67.2** | **46.7** |

## 4.2 Results on object detection and instance segmentation

**Setup.** We evaluate iFormer on the COCO object detection and instance segmentation tasks [31], where the models are trained on 118K images and evaluated on validation set with 5K images. Here, we use iFormer as the backbone in Mask R-CNN [75]. In the training phase, we use iFormer pretrained on ImageNet to initialize the detector, and adopt AdamW to train with an initial learning rate of $1 \times 10^{-4}$, a batch size of 16, and $1\times$ training schedule with 12 epochs. For training, the input

images are resized to be 800 pixels on the shorter side an no more than 1,333 pixels on the longer side. For the test image, its shorter side is fixed to 800 pixels. All experiments are implemented on mmdetection [78] codebase.

**Results.** Table 3 reports the box mAP ($AP^b$) and mask mAP ($AP^m$) of the compared models. Under similar computation configurations, iFormers outperforms all previous backbones. Specifically, compared with popular ResNet [47] backbones, our iFormer-S brings 8.2 points of $AP^b$ and 7.5 points $AP^m$ improvements over ResNet50. Compared with various Transformer backbones, our iFormers still maintain the performance superiority over their results. For example, our iFormer-B surpasses UniFormer-B [22], Swin-S [5] by 0.9 points of $AP^b$ and 3.5 points of $AP^b$ respectively.

## 4.3 Results on semantic segmentation

**Setup.** We further evaluate the generality of iFormer through a challenging scene parsing benchmark on semantic segmentation, *i.e.*, ADE20K [32]. The dataset contains 20K training images and 2K validation images. We adopt iFormer pretrained on ImageNet as the backbone of the Semantic FPN [79] framework. Following PVT [6] and UniFormer [22], we use AdamW with an initial learning rate of $2 \times 10^{-4}$ with cosine learning rate schedule to train 80k iterations. All experiments are implemented on mmsegmentation [80] codebase.

**Results.** In Table 4, we report the mIoU results of different backbones. On the Semantic FPN [79] framework, our iFormer consistently outperforms previous backbones on this task, in-

Table 4: Semantic segmentation with semantic FPN [79] on ADE20K [32]. The FLOPs are measured at resolution $512 \times 2048$.

| Method | #Param. (M) | FLOPs (G) | mIoU (%) |
|---|---|---|---|
| ResNet50 [47] | 29 | 183 | 36.7 |
| PVT-S [6] | 28 | 161 | 39.8 |
| TwinsP-S [76] | 28 | 162 | 44.3 |
| Twins-S [76] | 28 | 144 | 43.2 |
| Swin-T [5] | 32 | 182 | 41.5 |
| UniFormer-S$_{h32}$ [22] | 25 | 199 | 46.2 |
| UniFormer-S [22] | 25 | 247 | 46.6 |
| UniFormer-B [22] | 54 | 471 | 48.0 |
| **iFormer-S** | **24** | **181** | **48.6** |

cluding CNNs and (hybrid) ViTs. For instance, iFormer-S achieves 48.6 mIoU, surpassing UniFormer-S [22] by 2.0 mIoU, while using less computation complexity. Moreover, compared with UniFormer-B [22], our iFormer-S still achieves 0.6 mIoU improvement with only $1/2$ parameters and nearly $1/3$ FLOPs.

## 4.4 Ablation study and visualization

In this section, we conduct experiments to better understand iFormer. All the models are trained for 100 epochs on ImageNet, with the same training setting as described in Sec. 4.1.

**Inception token mixer.** The Inception mixer is proposed to augment the perception capability of ViTs in the frequency spectrum. To evaluate the effects of the components in the Inception mixer, we remove the max-pooling or convolution from the full model and then report the results in Table 5, where ✓ and ✗ denote whether or not the corresponding branch is enabled. Observably, combining attention with convolution and max-pooling can the highest classification accuracy. To further explore this scheme, Fig. 4 visualizes the Fourier spectrum of the Attention, MaxPool and DwConv branches

Table 5: Ablation study of Inception mixer and frequency ramp structure on ImageNet-1K. All the models are trained for 100 epochs.

| | Attention | MaxPool | DwConv | #Param. (M) | FLOPs (G) | Top-1(%) |
|---|---|---|---|---|---|---|
| Mixer | ✓ | ✓ | ✗ | 20 | 4.9 | 81.2 |
| | ✓ | ✗ | ✓ | 20 | 4.9 | 81.4 |
| | ✓ | ✓ | ✓ | 20 | 4.8 | 81.5 |
| Structure | $C_l/C \downarrow, C_h/C \uparrow$ | | | 19 | 4.7 | 80.5 |
| | $C_l/C = C_h/C$ | | | 19 | 4.7 | 80.7 |
| | $C_l/C \uparrow, C_h/C \downarrow$ | | | 20 | 4.8 | 81.2 |

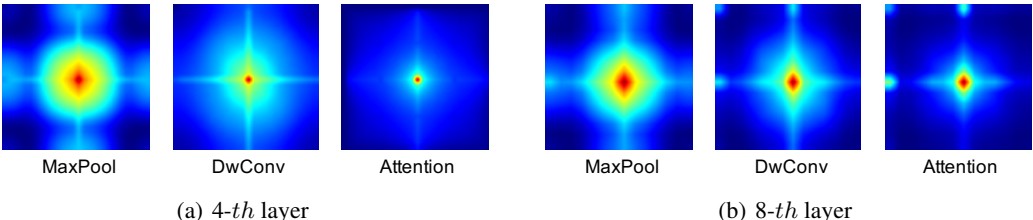

|                |                |                |                |                |                |
| :---: | :---: | :---: | :---: | :---: | :---: |
| MaxPool | DwConv | Attention | MaxPool | DwConv | Attention |
| (a) 4-*th* layer | | | (b) 8-*th* layer | | |

Figure 4: **(a) (b) Fourier spectrum of iFormer-S for the MaxPool, DwConv and Attention branches in the Inception mixer.** We can observe that attention mixer tends to reduce high-frequencies, while MaxPool and DwConv enhance them.

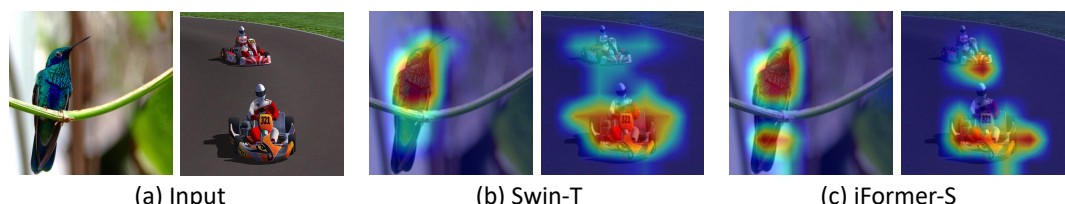

(a) Input          (b) Swin-T          (c) iFormer-S

Figure 5: **Grad-CAM [81] activation maps of Swin-T [5] and iFormer-S trained on ImageNet.**

in Inception mixer. We can see the attention mixer has higher concentrations on low frequencies; with the high-frequency mixer, *i.e.*, convolution and max-pooling, the model is encouraged to learn high frequency information. Overall, these results prove the effectiveness of the Inception mixer for expanding the perception capability of the Transformer in the frequency spectrum.

**Frequency ramp structure.** Previous investigations [27] show requirement of more local information at lower layers of the Transformer and more global information at higher layers. We accordingly assume that a frequency ramp structure, *i.e.*, decreasing dimensions at high-frequency components and increasing dimensions at low-frequency components from lower to higher layers, has a better trade-off between high-frequency and low-frequency components across all layers. In order to justify this hypothesis, we investigate the effects of the channel ratio ($\frac{C_h}{C}$ and $\frac{C_l}{C}$) in Table 5. It can be clearly seen that the model with $C_l/C \uparrow, C_h/C \downarrow$ outperforms the other two models, which is consistent with the previous investigations. Hence, this indicates the rationality of the frequency ramp structure and its potential for leaning discriminating vision representations.

**Visualization.** We visualize the Grad-CAM [81] activation maps of iFormer-S as well as Swin-T [5] models trained on ImageNet-1K in Fig. 5. It can be seen that compared with Swin, iFormer can more accurately and completely locate the objects. For example, in the hummingbird image, iFormer skips the branch and accurately attends to the whole bird including the tail.

## 5 Conclusion

In this paper, we present an Inception Transformer (iFormer), a novel and general Transformer backbone. iFormer adopts a channel splitting mechanism to simply and efficiently couple convolution/max-pooling and self-attention, giving more concentrations on high frequencies and expanding the perception capability of the Transformer in the frequency spectrum. Based on the flexible Inception token mixer, we further design a frequency ramp structure, enabling effective trade-off between high-frequency and low-frequency components across all layers. Extensive experiments show that iFormer outperforms representative vision Transformers on image classification, object detection and semantic segmentation, demonstrating the great potential of our iFormer to serve as a general-purpose backbone for computer vision. We hope this study will provide valuable insights for the community to design efficient and effective Transformer architectures.

**Limitation.** One obvious limitation of the proposed iFormer is that it requires manually defined channel ratio in the frequency ramp structure *i.e.*, $\frac{C_h}{C}$ and $\frac{C_l}{C}$ for each iFormer block, which needs rich experience to define better on different tasks. it is not trained on large scale datasets, *e.g.*,

ImageNet-21K [48], due to computational constraint, which will be explored in further. Also, iFormer requires manually defined channel ratio in the frequency ramp structure *i.e.*, $\frac{C_h}{C}$ and $\frac{C_l}{C}$ for each iFormer block, which needs rich experience to define better on different tasks. A straightforward solution would be to use neural architecture search.

## Acknowledgement

Weihao Yu would like to thank TRC program and GCP research credits for the support of partial computational resources. This project is in part supported by the National Research Foundation Singapore under its AI Singapore Programme (Award Number: AISG2-RP-2021-023).

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
