# A    Appendix

**Potential Impacts.** The study introduces a general vision Transformer, *i.e.*, iFormer, which can be used on different vision tasks, *e.g.*, image classification, object detection and semantic segmentation. iFormer has no direct negative societal impact. However, we will realize that iFormer as a general-purpose backbone can be used for harmful applications such as illegal face recognition.

## A.1    Results on semantic segmentation

**Setup.**    We further evaluate the generality of iFormer on semantic segmentation with the Uper-net [82] framework. Following the training settings in Swin [5], the model is trained for 160K iterations with a batch size of 16. For training, we use AdamW optimizer with an initial learning rate $6 \times 10^{-5}$. All experiments are implemented on mmsegmentation [80] codebase.

**Results.**    Table 6 shows the mIoU and MS mIoU results of different backbones based on the UperNet [82] framework. From these results, it can be seen that our iFormer achieves 48.4 mIoU and 48.8 MS mIoU, consistently surpassing previous backbones on this task. For instance, our iFormer-S outperforms the Swin-T [5] by 3.9 mIoU while using fewer parameters. Compered with UniFormer-S [22], iFormer-S still achieves 0.8 mIoU improvement.

Table 6: Semantic segmentation with semantic UperNet [82] on ADE20K [32]. The FLOPs are measured at resolution 512×2048.

| Method | #Param. (M) | FLOPs (G) | mIoU (%) | MS mIoU (%) |
|---|---|---|---|---|
| TwinsP-S [76] | 55 | 919 | 46.2 | 47.5 |
| Twins-S [76] | 54 | 901 | 46.2 | 47.1 |
| Swin-T [5] | 60 | 945 | 44.5 | 45.8 |
| Focal-T [64] | 62 | 998 | 45.8 | 47.0 |
| Shuffle-T [83] | 60 | 949 | 46.6 | 47.8 |
| UniFormer-S$_{h32}$ [22] | 52 | 955 | 47.0 | 48.5 |
| UniFormer-S [22] | 52 | 1008 | 47.6 | 48.5 |
| **iFormer-S** | **49** | **938** | **48.4** | **48.8** |

## A.2    Ablation study

### A.2.1    Down-sample and up-sample for self-attention

In this work, we propose the down- and up-sample structure to reduce the computational cost. Table 7 shows that, when removing this structure, iFormer-S has 7.0G FLOPs, and achieves 83.6% top-1 accuracy. However, by adopting the down- and up-sample structure, iFormer-S gets a similar accuracy (83.4%) but has much less FLOPs (4.8 G).

Table 7: Ablation study of down-up structure for Self-attention and kernel size of convolution.

| Former-S | Down- and Up | #Param. (M) | FLOPs (G) | Top-1(%) |
|---|---|---|---|---|
| | ✗ | 20 | 7.0 | 83.6 |
| | ✓ | 20 | 4.8 | 83.4 |

| Former-S | Kernel Size | | | Top-1(%) |
|---|---|---|---|---|
| | $5 \times 5$ | | | 83.2 |
| | $3 \times 3$ | | | 83.4 |

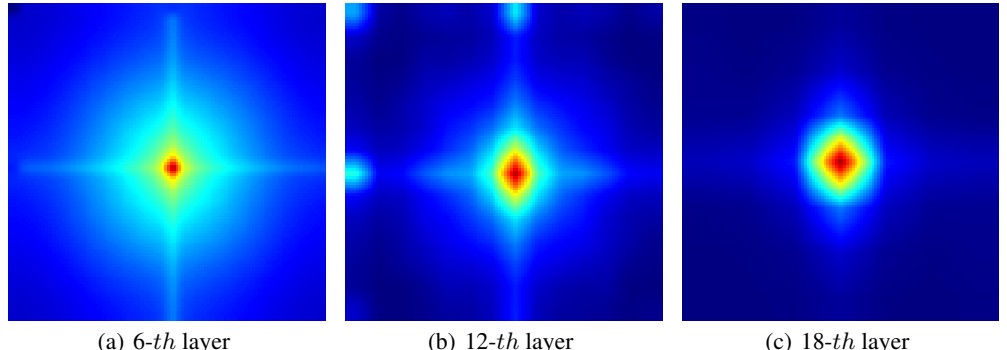

| (a) 6-*th* layer | (b) 12-*th* layer | (c) 18-*th* layer |

Figure 6: **(a) Fourier spectrum of 6-**$th$**, 12-**$th$ **and 18-**$th$ **layers for the iFormer-S.**

### A.2.2 The kernel size of convolution

We use large kernel (5×5) to replace $3 \times 3$ kernel of DWConv. The results are summarized in Tables 7. We can see that the model with $3 \times 3$ kernel achieves better results. We believe that this is because the self-attention has learned the information that is learnt by large-kernel DWConv. Besides, the smaller kernel is more conducive and effective for capturing high-frequency information.

### A.3 Visualization

### A.3.1 Fourier spectrum of different layers

Frequency ramp structure plays an important role in iFormer, which is designed to learn hierarchical representations, *i.e.*, more high-frequency signals at lower layers of the Transformer and more low-frequency signals at higher layers. In order to justify this hypothesis, we visualize the Fourier spectrum of feature maps for different iFormer layers in Fig 6. We can see that iFormer captures more high-frequency components at 6-$th$ layer and more low-frequency information at 18-$th$ layer. Moreover, the high-frequency information gradually decreases from 6-$th$ layer to 18-$th$ layer, and low-frequency information does the opposite. Hence, these results indicate our iFormer can effectively trade-off high- and low-frequency components across all layers.

### A.3.2 CAM

We futher show more examples of the Grad-CAM [81] activation maps of iFormer-S as well as Swin-T [5] models trained on ImageNet-1K in Fig. 7. These examples indicate that compared with Swin, iFormer can more accurately and completely attend to the key objects. Taking the hog picture as example, iFormer locates the hog accurately but Swin also locates irrelevant part.

### A.4 Configurations of iFormers

In this work, three variants of iFormer are used for a fair comparison under computation configurations, *i.e.*, iFormer-S, iFormer-B and iFormer-L. Table 8 shows their detailed configurations. Following Swin [5], iFormer adopts 4-stage architecture with $\frac{H}{4} \times \frac{W}{4}$, $\frac{H}{8} \times \frac{W}{8}$, $\frac{H}{16} \times \frac{W}{16}$, $\frac{H}{32} \times \frac{W}{32}$ input sizes, where $H$ and $W$ are the width and height of the input image. In each iFormer block, $C_h/C$ and $C_l/C$ are used to balance the high-frequency and low frequency components. As shown in Table 8, $C_h/C$ gradually decreases from shallow to deep layers, while $C_l/C$ gradually increases. iFormer block uses depthwise convolution and max-pooling as high-frequency mixers. We set the kernel sizes of depthwise convolution and max-pooling to $3 \times 3$.

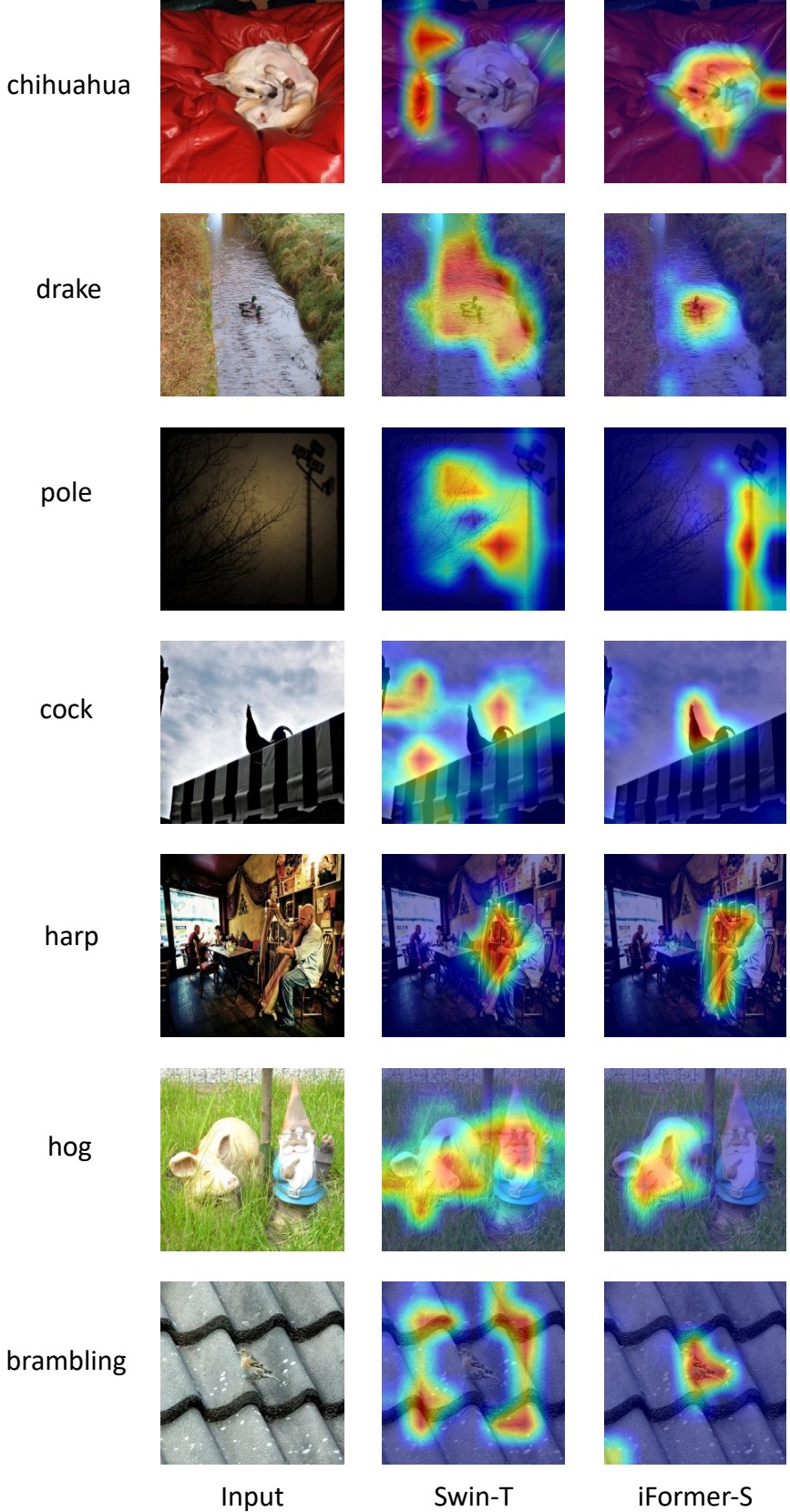

Figure 7: **Grad-CAM [81] activation maps of Swin-T [5] and iFormer-S trained on ImageNet.**

Table 8: Configurations of the variants of iFormer. Pool stride denotes the stride of the pooling and upsample layers in attention branch. The FLOPs are measured at resolution $224 \times 224$.

| Stage | Layer | iFormer-S | iFormer-B | iFormer-L |
|---|---|---|---|---|
| | Patch Embedding | $3 \times 3$, stride $2, 48$
$3 \times 3$, stride $2, 96$ | $3 \times 3$, stride $2, 48$
$3 \times 3$, stride $2, 96$ | $3 \times 3$, stride $2, 48$
$3 \times 3$, stride $2, 96$ |
| 1 | iFormer Block | $\begin{bmatrix} C_h/h = 2/3 \\ C_l/h = 1/3 \\ \text{pool stride } 2 \end{bmatrix} \times 3$ | $\begin{bmatrix} C_h/h = 2/3 \\ C_l/h = 1/3 \\ \text{pool stride } 2 \end{bmatrix} \times 4$ | $\begin{bmatrix} C_h/h = 2/3 \\ C_l/h = 1/3 \\ \text{pool stride } 2 \end{bmatrix} \times 4$ |
| | Patch Embedding | $2 \times 2$, stride $2, 192$ | $2 \times 2$, stride $2, 192$ | $2 \times 2$, stride $2, 192$ |
| 2 | iFormer Block | $\begin{bmatrix} C_h/h = 1/2 \\ C_l/h = 1/2 \\ \text{pool stride } 2 \end{bmatrix} \times 3$ | $\begin{bmatrix} C_h/h = 1/2 \\ C_l/h = 1/2 \\ \text{pool stride } 2 \end{bmatrix} \times 6$ | $\begin{bmatrix} C_h/h = 1/2 \\ C_l/h = 1/2 \\ \text{pool stride } 2 \end{bmatrix} \times 6$ |
| | Patch Embedding | $2 \times 2$, stride $2, 320$ | $2 \times 2$, stride $2, 384$ | $2 \times 2$, stride $2, 448$ |
| 3 | iFormer Block | $\begin{bmatrix} C_h/h = 3/10 \to 1/10 \\ C_l/h = 7/10 \to 9/10 \\ \text{pool stride } 1 \end{bmatrix} \times 9$ | $\begin{bmatrix} C_h/h = 4/12 \to 2/12 \\ C_l/h = 8/12 \to 10/12 \\ \text{pool stride } 1 \end{bmatrix} \times 14$ | $\begin{bmatrix} C_h/h = 4/14 \to 2/14 \\ C_l/h = 10/14 \to 12/14 \\ \text{pool stride } 1 \end{bmatrix} \times 18$ |
| | Patch Embedding | $2 \times 2$, stride $2, 384$ | $2 \times 2$, stride $2, 512$ | $2 \times 2$, stride $2, 640$ |
| 4 | iFormer Block | $\begin{bmatrix} C_h/h = 1/12 \\ C_l/h = 11/12 \\ \text{pool stride } 1 \end{bmatrix} \times 3$ | $\begin{bmatrix} C_h/h = 1/16 \\ C_l/h = 15/16 \\ \text{pool stride } 1 \end{bmatrix} \times 6$ | $\begin{bmatrix} C_h/h = 1/20 \\ C_l/h = 19/20 \\ \text{pool stride } 1 \end{bmatrix} \times 8$ |
| #Param. (M) | | 20 | 48 | 87 |
| FLOPs (G) | | 4.8 | 9.4 | 14.0 |