# OpenReview forum: "Inception Transformer"
_NeurIPS.cc/2022/Conference — NeurIPS 2022 Accept_

### Official Review · Reviewer_yYdE · 2022-07-06

**Rating:** 7
**Confidence:** 5
**Soundness:** 3 good
**Presentation:** 3 good
**Contribution:** 3 good

**Summary:**

This paper presents an iFormer architecture to capture high-frequency and low-frequency visual information. The proposed inception mixer module (i) splits the feature maps along the channel dimension and (ii) applies parallel convolution/max-pooling/self-attention paths to gather high-frequency and low-frequency information.
Besides, the authors also introduce a simple frequency ramp structure that adjusts the channel partitions across different layers. The experimental results look encouraging.

**Questions:**

Please carefully address the above-listed weaknesses. I will increase the ratings if the proposed method shows advantages under either the TINY or  LARGE configuration.

> Update:

After reading the comments of other reviewers and the authors' responses carefully, I tend to increase my ratings due to the potential high impact on the community.

**Ethics Review Area:**

["I don’t know"]

**Limitations:**

Yes.

**Strengths And Weaknesses:**

> Strengths

👍 The motivation is clear and the presented idea is very simple and easy to follow.

👍 The proposed iFormer shows slightly better performance while enjoying smaller GFLOPs.

> Weaknesses

👎 The proposed method is TOO NAIVE & lacks NOVELTY. According to Figure 3 and Figure 4, we can see that these three different branches capture information of different frequencies. However, the default **multi-head self-attention is capable to learn such information given enough training epochs** and previous efforts ([ConVit, ICML 2021](https://arxiv.org/abs/2103.10697), [EarlyConvolutions, NIPS 2021](https://proceedings.neurips.cc/paper/2021/file/ff1418e8cc993fe8abcfe3ce2003e5c5-Paper.pdf), [MetaFormer, CVPR 2022](https://openaccess.thecvf.com/content/CVPR2022/papers/Yu_MetaFormer_Is_Actually_What_You_Need_for_Vision_CVPR_2022_paper.pdf))have shown that **injecting the inductive biases with either convolution or max-pooling show better training efficiency**.

👎 The proposed method does not provide any results under LARGE configurations. The authors should justify the position of the proposed iFormer on either a TINY configuration that pursues faster running speed & better efficiency or a LARGE configuration that pursues higher accuracy. However, the authors fail to show the proposed method is promising in both aspects. Therefore, it is necessary to present the comparisons to either [Mobile-Former, CVPR 2022](https://arxiv.org/pdf/2108.05895.pdf)/[MobileViT, ICLR 2022](https://openreview.net/pdf?id=vh-0sUt8HlG) or Swin-L/Focal-L/CSwin-L under the fair settings instead of only on an intermediate setting that the community does not care.

👎 The authors should include the results with the combination of ``Attention and DWConv''. According to our previous experience, we can apply large kernel DWConv to achieve better performance than the naive MaxPool operation.

---

> ### Author Response · Authors · 2022-08-02
> **Response to Reviewer yYdE**
>
> We thank the reviewer for the insightful and constructive comments. Please find the response to your questions below:
>
> **Q1: The proposed method does not provide any results under LARGE configurations. It is necessary to present the comparisons to either Mobile-Former, CVPR 2022/MobileViT, ICLR 2022 or Swin-L/Focal-L/CSwin-L under the fair settings instead of only on an intermediate setting.**
>
> **Authors’ reply:** Thanks.  Due to the limited rebuttal time, we only completed the comparison experiment of a tiny version of the proposed iFormer with Mobile-Former under similar parameters. iFormer-Tiny with 10.3M parameters achieves 80.2\% accuracy, and improves over the accuracy 79.3\% of Mobile-Former with 14M parameters by 0.9\%. Considering the large computation cost, we will add the results of large configuration for comparisons in a future version.
>
> In this paper, our goal is to disclose the defects of transformer from the perspective of the frequency domain and propose a simple and effective solution. We hope that while pursuing higher accuracy, this study will provide valuable insights for the community to design efficient and effective Transformer architectures.
>
> **Q2: The default multi-head self-attention is capable to learn different frequencies given enough training epochs. And previous efforts have shown inductive biases with either convolution or max-pooling show better training efficiency.**
>
> **Authors’ reply:** Thanks. Self-attention can indeed learn different frequencies given enough training, but it is highly capable of capturing low-frequencies in the visual data while not very powerful for learning high-frequencies, which is exactly what we claim in this paper. To prove this point, we visualize the Fourier spectrum of ViT or self-attention in Fig. 1 and Fig. 4 of the submitted manuscript. It is obvious that the Fourier spectrum of ViT or self-attention contains both high- and low-frequency information but the low-frequency information dominates.
>
> Unlike ConVit, EarlyConvolutions and MetaFormer stacking convolution and attention layers in a serial manner, iFormer uses an inception architecture to combine self-attention with convolution and max-pooling, simply and efficiently helping it capture more high frequencies and expanding the perception capability of the Transformer in the frequency spectrum. Moreover, the pioneering frequency ramp structure also proposed in our work enables an effective trade-off between high- and low-frequency components across all layers. Actually, the differences between iFormer and previous hybrid methods have been discussed in the "Introduction" and "Related Work" parts of the submitted manuscript. In experiments, we have also compared iFormer with these hybrid architectures on ImageNet-1K.
>
> **Q3: The authors should include the results with the combination of ``Attention and DWConv''. According to our previous experience, we can apply large kernel DWConv to achieve better performance than the naive MaxPool operation.**
>
> **Authors’ reply:** As per your suggestion, we conduct ablation study by removing each component from the network. We can see that iFormer-S with convolution, max-pooling and attention outperforms the combination of convolution and attention by 0.1\%. Moreover, we use large kernel (5x5) to replace 3x3 kernel of DWConv. The results are summarized in the following Tables. We can see that the model with 3x3 kernel achieves better results. We believe that this is because the self-attention has learned the information that is learnt by large-kernel DWConv. Besides, the smaller kernel is more conducive and effective for capturing high-frequency information.
>
> | Attention | MaxPool | DwConv | Top-1(%, 100epoch) |
> |:---------:|:-------:|:------:|:--------:|
> |      &#10004;      |    &#10004;     |      &cross;     |   81.2   |
> |      &#10004;      |    &cross;        |     &#10004;   |   81.4   |
> |      &#10004;      |     &#10004;    |     &#10004;   |   81.5   |
>
>
> | Kernel | Top-1(%, 300epoch)  |
> |:------:|:-----------------:|
> |   5x5  |        83.2       |
> |   3x3  |        83.4       |

---

### Official Review · Reviewer_S6Nf · 2022-07-10

**Rating:** 8
**Confidence:** 5
**Soundness:** 4 excellent
**Presentation:** 4 excellent
**Contribution:** 3 good

**Summary:**

This paper proposes Inception Transformer (iFormer) to enhance the low-pass filters in standard transformers. Specifically, the authors introduce an Inception mixer that has parallel convolution/max-pooling path and self-attention path, capturing high- and low-frequency information respectively. The Inception mixer differs from existing parallel CNN/attention hybrids in two aspects: 1) Unlike [23, 26, 47] that process all channels in each branch, the Inception mixer adopts a channel splitting mechanism to reduce information redundancy. 2) The Inception mixer also contains a fusion module to merge the outputs from low- and high-frequency branches, instead of simple feature concatenation.

Another novelty of the proposed iFormer is the frequency ramp structure. It trade-offs high- and low-frequency info across different layers by gradually decreasing the channel dimensions fed to the high-frequency mixer and increasing those to the low-frequency mixer.

The full iFormer achieves SOTA performance on a series of vision tasks, including image classification, object detection and segmentation. It shows better performance-cost trade-off than representative ViTs, CNNs and their hybrid variants, showing the potential to serve as a general transformer backbone.

**Questions:**

- In the low-frequency mixer, multi-head self-attention is made more efficient by using average pooling first and then upsampling. How much saving is obtained in compute? Will this hurt the attention performance too much (although I get the intention is to focus on embedding global information in the attention branch)?
- What's the exact policy to change the channel ratio across layers? It's linear? How do different policies perform (or do they perform differently at all)?
- How does the feature fusion module compares to direct concatenation [47] in performance?

**Limitations:**

Both the limitations and potential negative societal impact are given. It makes sense to bring up the limitation that iFormer requires manually defined channel ratio in the frequency ramp structure.

**Strengths And Weaknesses:**

Strengths:
+ It's new to consider enhancing the capability of transformers to capture high-frequency information.
+ The two key novelties--Inception mixer and frequency ramp structure--are intuitive and make sense. There are ablations provided to support their positive roles.
+ Although the Inception mixer has a parallel structure that is similarly adopted by existing papers, there are still technical differences (ie. channel splitting and fusion module) that prove useful.
+ Strong performance is obtained on various vision tasks, and that comes without sacrificing model efficiency.

Weaknesses:
- The paper should benefit from a detailed quantification of the high- and low-frequency information (e.g. via Fourier analysis) captured for each layer. Is there any interesting pattern across layers? This is the core contribution of the whole paper, but the analysis about how Inception mixer fulfills the task in its low- and high-frequency branches is missing (example qualitative visualization in Fig. 4 may not be enough).
- As mentioned in the limitations, the channel splitting scheme is lacking analysis and/or ablations.

---

> ### Author Response · Authors · 2022-08-02
> **Response to Reviewer S6Nf**
>
> We thank the reviewer for the insightful and constructive comments. Please find the response to your questions below:
>
> **Q1: What’s the exact policy to change the channel ratio across layers? It’s linear? How do different policies perform (or do they perform differently at all)?**
>
> **Authors’ reply:**  According to reference paper [59] as indexed in our submission, the dimension of each attention head is set to 32. We use the head number to compute the channel ratio $C_l / C$. Specifically, for iFormer-S, its four blocks respectively have channel dimensions $C=$ 96, 192, 320, 384 and attention head number 3, 6, 10 and 12. Then we  approximately linearly scale $C_l / C$ by setting it as $1/3 (=20/60)$ in the $1$st bock, $1/2 (=30/60)$ in the $2$nd bock, $7/10 (=42/60)$ and $9/10 (=54/60)$ in the $3$rd bock, $11/12 (=55/60)$ in the $4$th bock. In addition to linear scaling, we have also tried cosine scaling to increase the channel ratio of low-frequencies, and it achieves 83.4\% accuracy on iFormer-S which is the same accuracy as that of linear scaling. In addition to the manually defined strategies in the frequency ramp structure, neural architecture search can be used to automatically search for a balance ratio between the high- and low-frequency components, which is left as our future work.
>
> **Q2: In the low-frequency mixer, multi-head self-attention is made more efficient by using average pooling first and then upsampling. How much saving is obtained in compute? Will this hurt the attention performance too much**
>
> **Authors’ reply:**  In this work, we propose the down- and up-sample structure to reduce the computational cost. When removing this structure, iFormer-S has 7.0G FLOPs, and achieves 83.6\% top-1 accuracy. However, by adopting the  down- and up-sample structure, iFormer-S gets a similar accuracy (83.4\%) but has much fewer FLOPs (4.8 G).
>
> **Q3: How does the feature fusion module compares to direct concatenation [47] in performance?**
>
> **Authors’ reply:**  Our feature fusion module for iFormer-S achieves 83.4\% Top-1 accuracy, while the result of direct concatenation [47] is 83.0\%.  As the upsample operation results in excessive smoothness between adjacent tokens, we use a depthwise convolution to exchange information between patches, which elegantly overcomes this issue.
>
> **Q4: The paper should benefit from a detailed quantification of the high- and low-frequency information.**
>
> **Authors’ reply:**  \paragraph{Authors’ reply:} Thanks for your suggestion. In reference works [19,20], they analyzed the frequency via visualizing the Fourier spectrum. Following [19,20], we also visualize the Fourier spectrum of feature maps for different iFormer layers in Fig. 6 in the submitted manuscript. The visualizations show that our iFormer can effectively trade-off high- and low-frequency components across all layers. For better understanding how the Inception mixer fulfills the task in its low- and high-frequency branches, we will try more methods to further analyze the frequency information in the final version.

---

### Official Review · Reviewer_Bzyx · 2022-07-11

**Rating:** 8
**Confidence:** 4
**Soundness:** 4 excellent
**Presentation:** 3 good
**Contribution:** 4 excellent

**Summary:**

This paper designs a new family of multi-branched visual transformer models, called Inception Transformer. Motivated by the spectrum properties of local operations (convolution & pool) and long-range operations (self-attention), a multi-branched Inception Mixer block is proposed to capture both high/low frequency information from the images. A channel splitting mechanism is applied to adopt hybrid operation at each stage.
In addition, the author assumes that low layers of network favor local/high-frequency pattern and high layers prefer global/low-frequency pattern. Therefore, a frequency ramp structure is utilized to increase the attention dimensions and shrink the convolution dimensions as the network goes deeper.
Comprehensive evaluation on image classification, object detection, instance segmentation and semantic show strong performance.

**Questions:**

1.	I hope the author should provide more discussions on the core difference between iFormer and previous multi-branched network structure.
2.	Frequency ramp strategy. In the “Frequency ramp structure” section, no description on the how to increase the channel ratio C_l/C. Is it linear scaling according to the depth or exponential increasing? I see the channel number in the appendix, but it seems to be a “magic number” that I am not sure how did you decide on the current number. Moreover, an ablation on how to select the channel number should be included in the experimental part.
3.	Down-sample and Up-sample Self-attention. To reduce the computational cost, the author proposes in Equation 7 to (1) down-sample the feature map with average pooling (2) Do self-attention (3) Up-sample to the full size. This operation is not mentioned in previous studies. I am curious about its computational complexity and performance, with the vanilla ViT structure.

**Limitations:**

Please put more discussions on societal impact.

**Strengths And Weaknesses:**

Overall, I enjoy reading this paper, with straightforward motivation and clear writing.
Strengths:
1.	Strong motivation. As the self-attention operation captures the long-range dependency (shown in Figure 1(a)(b)), a local branch with convolution and pooling operations is well-motivated to compensate the frequency preference of attention operation.
2.	Simple and efficient structure. Instead of applying different operations on the same feature maps, the high/low frequency mixer is applied on different channel slices in a parallel manner. Also, the structure that gradually increases the global operation channel makes sense to me
3.	Promising empirical results. On ImageNet, iFormer-S/iFormer-B surpass the current best performed architecture by 0.5 %/0.7% in accuracy on 224 resolution. The iFormer-B achieves 48.3 mAP on COCO2017 val and 48.6 mIOU on ADE20K. All results are surprisingly well.

Weakness:
1.	Lack of discussion with previous multi-branched network structure. Since the current paper focus on the design of a multi-branched block structure, a detailed comparison and discussion should be included, like ResNeXt[1], GoogleNet Family and Inception Family. Although they did not include self-attention in their branches, some of the networks do have the mixed kernel size (1x1 Vs 5x5), which is also another form of low/high frequency mixer. Also, the multi-branched network seems to have better generalization and optimization properties [2] compared with the single-branched counter parts. A simple citation is insufficient to highlight your improvement over them. In addition, as the paper is largely driven by the low/high pass filter design in signal processing, I would encourage the author incudes some reference on this topic.
2.	Problems on ablation study. To be scientific, ablation study means quantifying the influence of each modular design by removal/changing of one component from the entire system. But in table 5 row 1-4, the authors are accurately adding one component at each time. I know that a lot of paper take this style of ablation study, but scientifically, this is not a valid ablation study. Better fix this.

[1] Aggregated Residual Transformations for Deep Neural Networks (CVPR 2017)
[2] Deep Neural Networks with Multi-Branch Architectures Are Intrinsically Less Non-Convex (ICML 2019)

---

> ### Author Response · Authors · 2022-08-02
> **Response to Reviewer Bzyx**
>
> We thank the reviewer for the insightful and constructive comments. Please find the response to your questions below:
>
> **Q1: I hope the author should provide more discussions on the core difference between iFormer and previous multi-branched network structure.**
>
> **Authors’ reply:** Thanks for your suggestion. The main differences between iFormer and the ResNeXt/Inception-alike CNN family are as follows:
> 1) From the motivation, this work aims to disclose the problem of vanilla ViTs, and provides a solution to augment the perception capability of ViTs in the frequency spectrum via a multi-branched structure, while ResNeXt/Inception aims to improve the effectiveness and efficiency of CNNs.
> 2) The proposed iFormer uses the multi-branched structure to flexibly balance the different frequencies in the representation so that the frequency ramp structure can effectively trade-off high- and low-frequency components across all layers. In contrast, the RexNeXt/Inception-alike CNN family does not consider the balance between different branches.
> 3) For the inception token mixer, we simply use convolution and max-pooling as the high-frequency mixer, which can be replaced if the research community finds any other better token mixer to learn high-frequency information. Differently, the branches of RexNeXt/Inception-alike CNN family are fixed.
>
> **Q2: In addition, as the paper is largely driven by the low/high pass filter design in signal processing, I would encourage the author includes some references on this topic.**
>
> **Authors’ reply:** Thanks for your suggestion. We will add some references about this topic in the final version.
>
> **Q3: Problems on ablation study.**
>
> **Authors’ reply:** Thanks for your valuable suggestion. As per your suggestion, we conduct ablation study by removing the component (max-pooling or convolution) from the inception mixer. As seen from the following Table, combining attention with convolution and max-pooling can achieve the highest classification accuracy on ImageNet.
> | Attention | MaxPool | DwConv | Top-1(%) |
> |:---------:|:-------:|:------:|:--------:|
> |      &#10004;      |    &#10004;     |      &cross;     |   81.2   |
> |      &#10004;      |    &cross;        |     &#10004;   |   81.4   |
> |      &#10004;      |     &#10004;    |     &#10004;   |   81.5   |
>
> **Q4: How to increase the channel ratio $C_l / C$**
>
> **Authors’ reply:** According to reference paper [59] as indexed in our submission, the dimension of each attention head is set to 32. We use the head number to compute the channel ratio $C_l / C$. Specifically, for iFormer-S, its four blocks respectively have channel dimensions $C=$ 96, 192, 320, 384 and attention head number 3, 6, 10 and 12. Then we  approximately linearly scale $C_l / C$ by setting it as $1/3 (=20/60)$ in the $1$st bock, $1/2 (=30/60)$ in the $2$nd bock, $7/10 (=42/60)$ and $9/10 (=54/60)$ in the $3$rd bock, $11/12 (=55/60)$ in the $4$th bock.
>
> **Q5: Down-sample and Up-sample Self-attention.**
>
> **Authors’ reply:** In this work, we propose the down- and up-sample structure to reduce the computational cost. When removing this structure, iFormer-S has 7.0G FLOPs, and achieves 83.6\% top-1 accuracy. However, by adopting the down- and up-sample structure, iFormer-S gets a similar accuracy (83.4\%) but has much less FLOPs (4.8 G).

---

### Official Review · Reviewer_X9JV · 2022-07-12

**Rating:** 7
**Confidence:** 4
**Soundness:** 4 excellent
**Presentation:** 3 good
**Contribution:** 4 excellent

**Summary:**

In this paper, the authors propose a novel iFormer architecture to capture both the high and low frequencies in visual data. Motivated by the observation that ViT tends to capture few high-frequency signals (i.e., showing the characteristics of low-pass filters), the authors propose an Inception token mixer, comprising high- and low-frequency mixers to extract the corresponding frequency information on the feature channels. Also, derived from the finding that lower and higher layers generally need more local and global information, respectively, the authors devise a frequency ramp structure to trade off different-frequency components across all the layers. Experiments across various tasks including classification, detection, and segmentation, convincingly validate the effectiveness of the proposed iFormer with detailed explanations and discussions。

**Questions:**

- Have you tried other balancing strategies other than the manner of setting the sum of the ratios to 1 in the proposed frequency ramp structure?
- Can you share some insights on the future work that can be derived from this work?
- Do you plan to make your code publicly available for follow-up research?



**Limitations:**

Yes, limitations have been discussed. I do not find the potential negative societal impact of this work.

**Strengths And Weaknesses:**

[Strengths]
- The proposed iFormer is very well-motivated. Deriving from the results of the Fourier spectrum and the relative log Fourier amplitudes, the authors accordingly develop the iFormer backbone to enhance the perception capability of the Transformer in the frequency spectrum.
- The experiments are comprehensive and convincing. Results across three challenging tasks show that the proposed iFormer outperforms existing Transformer architecture. Extensive ablation studies and visualizations also demonstrate the effectiveness of the devised two key modules of the Inception token mixer and the frequency ramp structure.
- The paper is well-written and easy to follow. I enjoy the writing style that starts from the detailed discussions and analysis on the derived observations and then accordingly elaborates the proposed method.

[Weaknesses]
- The authors are suggested to denote the symbols like X_{h1}, X_{h2}, Y_{h1}, and Y_{h2} in Fig. 3, which will make it much easier to relate Fig. 3 back to the text.
- The design of a frequency ramp structure that uses a channel ratio to balance the high- and low-frequency components seems straightforward. I wonder if the authors have tried other designs for frequency information balancing and how they perform.
- It will be great if the authors could share some insights on the future work that can be derived from this work.
- The punctuations for Eq. 1 and Eq. 5 are missing.

---

> ### Author Response · Authors · 2022-08-02
> **Response to Reviewer X9JV**
>
> We thank the reviewer for the insightful and constructive comments. Please find the response to your questions below:
>
> **Q:  Have you tried other balancing strategies in the proposed frequency ramp structure?**
>
> **Authors’ reply:** As shown in Table 5, we have tried three balancing strategies, including increasing, invariant and decreasing structures for the low-frequency component. The results reveal that the increasing structure outperforms the other two structures. For the increasing structure, we linearly increase the channel dimensions to low-frequency mixer. In addition, we have also tried cosine scaling to increase the channel ratio of low-frequencies, achieving 83.4% accuracy on iFormer-S, the same accuracy as that of using linear scaling. It is worth mentioning that beyond the manually defined strategies in the frequency ramp structure, neural architecture search can be used to automatically search for a balance ratio between high- and low-frequency components, which is left as our future work.
>
> **Q2: Can you share some insights on the future work that can be derived from this work?**
>
> **Authors’ reply:** This work mainly shows that well-balancing the low- and high-frequency information in data can benefit the performance on many tasks, e.g. classification and detection. Accordingly, any solution towards better learning of low- and high-frequency information should be expected for better performance. In our opinion, there are at least three directions to explore.
> 1) In this work, we simply apply the convolution operation and max-pooling to learn the high-frequency components with a parallel structure. In the future, a further optimized architecture of high-frequency token mixer will be explored for capturing high-frequency representations.
> 2) Human visual system extracts visual elementary features at different frequencies and interacts the information between different frequencies. This gives us an inspiration that mutually interacting high- and low-frequency representations may benefit the performance.
> 3) It has been shown that the adaptive attention has the ability to capture both high- and low-frequencies. We then consider using a regularization method to help improve the ability of attention to learn high-frequency information.
>
> **Q3: Do you plan to make your code publicly available for follow-up research?**
>
> **Authors’ reply:**  Yes, we will release the code and models soon.
>
> **Q4: Denote the symbols in Fig.3 and the punctuations for Eq.1 and Eq.5 are missing.**
>
> **Authors’ reply:** Thanks for your careful proofreading. We will revise these typos and also further refine the paper in the final version.

---

### Meta-Review · Area_Chair_n7LM · 2022-08-25

**Recommendation:** Accept
**Confidence:** Certain

**Metareview:**

This paper proposes a novel multi-branch style architecture for vision tasks, motivated by a frequency perspective of deep network behaviors. All reviewers are very positive about the motivation, presentation and experimental results. The AC believes this should be a good contribution to the neural architecture design community.

**Award:**

No

---

### Decision · Program_Chairs · 2022-09-14

Accept